# All-*trans* retinoic acid downregulates HBx levels via E6-associated protein-mediated proteasomal degradation to suppress hepatitis B virus replication

Jiwoo Han[1], Kyung Lib Jang [1,2,3]*

**1** Department of Integrated Biological Science, The Graduate School, Pusan National University, Busan, Republic of Korea, **2** Department of Microbiology, College of Natural Science, Pusan National University, Busan, Republic of Korea, **3** Microbiological Resource Research Institute, Pusan National University, Busan, Republic of Korea

* kljang@pusan.ac.kr

**Data Availability Statement:** All relevant data are within the manuscript and its Supporting Information files.

## Abstract

All-*trans* retinoic acid (ATRA), recognized as the principal and most biologically potent metabolite of vitamin A, has been identified for its inhibitory effects on hepatitis B virus (HBV) replication. Nevertheless, the underlying mechanism remains elusive. The present study reveals that ATRA induces E6-associated protein (E6AP)-mediated proteasomal degradation of HBx to suppress HBV replication in human hepatoma cells in a p53-dependent pathway. For this effect, ATRA induced promoter hypomethylation of E6AP in the presence of HBx, which resulted in the upregulation of E6AP levels in HepG2 but not in Hep3B cells, emphasizing the p53-dependent nature of this effect. As a consequence, ATRA augmented the interaction between E6AP and HBx, resulting in substantial ubiquitination of HBx and consequent reduction in HBx protein levels in both the HBx overexpression system and the *in vitro* HBV replication model. Additionally, the knockdown of E6AP under ATRA treatment reduced the interaction between HBx and E6AP and decreased the ubiquitin-dependent proteasomal degradation of HBx, which prompted a recovery of HBV replication in the presence of ATRA, as confirmed by increased levels of intracellular HBV proteins and secreted HBV levels. This study not only contributes to the understanding of the complex interactions between ATRA, p53, E6AP, and HBx but also provides an academic basis for the clinical employment of ATRA in the treatment of HBV infection.

## Introduction

Retinoids, classified as derivatives and analogs of vitamin A, play an important role in a broad range of biological activities including vision, developmental processes, including cellular differentiation, metabolism, and the maintenance of cellular homeostasis [1]; thus, they have garnered significant interest across various areas of pharmaceutical research. Notably, all-*trans* retinoic acid (ATRA) initially found applications in dermatology for treating ichthyosis and managing conditions like psoriasis and acne. Its versatile capabilities extend to diverse

**Funding:** This work was supported by a National Research Foundation of Korea (NRF) grant funded by the Korean government (MEST) (NRF-2019R1A2C2011478) (KLJ). The URL for NRF is https://www.nrf.re.kr/index. The funders had no role in study design, data collection and analysis, decision to publish, or preparation of the manuscript.

**Competing interests:** No authors have competing interests.

physiological systems, contributing to the prevention and treatment of skin aging, allergic respiratory conditions, and metabolic disorders [2, 3]. Moreover, ATRA has gained approval for its use in combination with other drugs in cancer therapy, notably acute promyelocytic leukemia (APL) [4, 5]. Recent studies have also highlighted the antiviral properties of ATRA, demonstrating its effectiveness against various viruses, including hepatitis B virus (HBV) human immunodeficiency virus type 1, hepatitis C virus, human papillomavirus type 16 (HPV-16), measles virus, and severe acute respiratory syndrome coronavirus 2 [6–12]. Despite the increasing evidence on the antiviral effects of ATRA, the precise action mechanisms remain poorly understood.

The involvement of HBV is significant in the progression of liver disorders in humans, including hepatitis, hepatocellular carcinoma (HCC), and liver cirrhosis [13, 14]. HBV belongs to the *Hepadnaviridae* family and it undertakes replication and encapsidation of a partially double-stranded circular DNA genome, roughly 3.2 kilobase pairs in length, using the reverse transcription of a pregenomic RNA [13, 15]. The shortest gene within the four open reading frames encodes a 17-kDa multifunctional protein known as HBV X protein (HBx). HBx has been strongly involved in HCC development due to its roles in regulating cellular signaling pathways, activating cellular genes transcriptionally, and dysregulating cell growth, apoptosis, and lipid metabolism [14, 16]. Additionally, HBx serves as a positive regulator of HBV replication by directly stimulating the four viral promoters to synthesize HBV mRNA and pre-genomic RNA out of a covalently closed circular DNA template [13, 17, 18]. While there is substantial evidence concerning the involvement of HBx in HBV replication, the exact mechanism by which HBx is regulated throughout the process of HBV replication still remains elusive.

Previous reports have presented evidence of the divergent effects of HBx and ATRA on the gene expression engaged in cellular growth regulation. Notably, HBx is known to stimulate cell growth by suppressing the expression of p21 and p16 [19, 20], while ATRA suppresses cellular proliferation by elevating the expression levels of p14, p16, and p21 [21, 22]. This dualistic relationship encompasses the reciprocal antagonism between HBx and ATRA concerning the expression levels of p14, p16, and p21. This, in turn, affects the regulatory mechanisms governing cell growth [23–25]. Additionally, the depressant effect of ATRA on HBV replication, which was demonstrated in previous studies [10, 11], may represent another facet of the interaction between HBx and ATRA. Nevertheless, comprehending the intricate mechanism behind this impact has proven challenging, primarily attributed to the obstacles in culturing HBV in cells. Recent progress in *in vitro* HBV infection and replication systems has partially alleviated these challenges [26].

According to previous reports, ATRA has been shown to activate p53 by increasing its transcriptional activity or upregulating its protein levels via stabilization [21, 27, 28], although the detailed mechanisms are still unknown. Additionally, the antiviral attributes of p53 opposed to HBV have been validated by introducing ectopic p53 expression in cell culture models of HBV replication, leading to the suppression of HBV replication [29, 30]. In this study, we employed an *in vitro* system for HBV replication that has been optimized to facilitate robust HBV replication in cell culture systems [31]. This platform enabled the explore the role of ATRA in negatively regulating HBV replication. Our investigation commenced by examining whether ATRA downregulates HBV replication through the engagement of p53. Subsequently, we explored the significance of HBx in the p53-dependent suppression of HBV replication induced by ATRA Furthermore, we investigated whether and how ATRA increases the proteasomal degradation of HBx in both HBx overexpression and *in vitro* HBV infection systems. Ultimately, our aim was to establish that ATRA suppresses HBV replication by reducing HBx levels through E6-associated protein (E6AP)-mediated proteasomal degradation in a p53-dependent pathway.

## Materials and methods

### Cell culture

Dulbecco's modified Eagle's medium (DMEM; Welgene, Gyeongsan, Republic of Korea, Cat No. LM00105) was used to grow cells including 10% (v/v) fetal bovine serum (FBS; Capricorn Scientific, Ebsdorfergrund, Germany, Cat No. FBS-22A), penicillin G 100 units (Sigma-Aldrich, Cat No. A1720) per ml and streptomycin 100 μg (United States Biological, Salem, MA, USA, Cat No. 21865) per ml, in 5% $CO_2$-humidified atmosphere at 37˚C. The human HCC cell line, Hep3B (Cat No. 88064), and HepG2 (Cat No. 88065) were purchased from the Korean Cell Line Bank (KCLB, Seoul, Republic of Korea), respectively. Hep3B-NTCP and HepG2-NTCP were established by transfection with RC210241, using 500 $μg·mL^{-1}$ G418 sulfate selection marker (Sigma-Aldrich, St. Louis, MO, USA, Cat No. A1720). in 6-well plate, $2×10^5$ cells were transfected with transfection reagent TurboFect (Thermo Fisher Scientific, Waltham, MA, USA, Cat No. R0532) for transient expression. In the case of treatment of chemical compounds, cells were treated with MG132 (Millipore, Burlington, MA, USA, Cat No. 474790) and cycloheximide (CHX; Sigma-Aldrich, Cat No. C7698) under the indicated conditions. Cells were incubated in a medium containing ATRA (Sigma-Aldrich, Cat No. R2625) for 4 hours before harvesting.

### Plasmids

Plasmid CMV-3 HA1-HBX3, the HBx expression plasmid encodes HBx downstream of three copies of the influenza virus haemagglutinin (HA) epitope (YPYDVPDYA) was used which is described before [19]. The 1.2- mer-WT HBV replicon including 1.2 units of the genotype D HBV genome and its HBx-null (1.2mer HBx-null) counterpart was used as described before [32]. Plasmid cMVT N-HA-hE6AP with human HA-tagged E6AP (amino acids 262–853) and pCH110 which encodes the *Escherichia coli* β-galactosidase gene were acquired from Addgene (Watertown, MA, USA). The plasmid RC210241 (Cat No. 003049), including the human Na+ -taurocholate co-transporting polypeptide (NTCP), was purchased from OriGene (Rockville, MD, USA). Scrambled (SC) shRNA (Cat No. sc-7007), p53 shRNA (Cat No. sc-29435), and E6AP shRNA plasmids were purchased from Santa Cruz Biotechnology (Santa Cruz, CA, USA). pHA-ubiquitin (Ub) and pCMV p53-wild-type (WT) were generously furnished by Dr. Y. Xiong (UNC-Chapel Hill, NC, USA) and Dr. C.-W. Lee (Sungkyunkwan University, Suwon, Republic of Korea), respectively.

### HBV cell culture system

For HBV stock, Hep3B-NTCP cells were transiently or stably transfected with the 1.2-mer HBV replicon plasmid for 48 hours [33]. Infection of HBV was carried out in 6-well plates with a multiplicity of genome equivalents (GEQ) set at 50 for a duration of 4 days, following a refined HBV cell culture system with minor adjustments [31, 34]. Shortly, $2 × 10^5$ cells were exposed to $10^7$ GEQ of HBV for 24 h in DMEM including 4% polyethylene glycol 8000 (PEG 8000, Sigma-Aldrich, Cat No. D4463), 3% FBS, and 2% dimethyl sulfoxide (DMSO, Sigma-Aldrich, Cat No. D8418). Subsequently, after two washes with serum-free DMEM, the cells were cultured in DMEM including 4% PEG 8000, 3% FBS, and 2% DMSO for an extra 3 days.

### HBV DNA quantitative real-time PCR

Quantification of extracellular HBV DNA concentrations was conducted using qPCR, following established procedures demonstrated earlier [31, 35]. In brief, extraction of HBV DNA acquired from the cell supernatant was performed with the QIAamp DNA Mini Kit (Qiagen,

Hilden, Germany, Cat No. 51306). For standard PCR examination, amplification of HBV genomic DNA was carried out with 2× Taq PCR Master Mix 1 (BioFACT, Daejeon, Republic of Korea, Cat No. ST301-19h). HBV 1399F (5'-TGG TAC CTC CGC GGG ACG TCC TT-3') and HBV 1632R (5'-AGC TAG CGT TCA CGG TGG TCT CC-3') were used as primer pairs. In qPCR analysis, amplification of HBV DNA was conducted with the SYBR Premix Ex Taq II (Takara Bio, Shiga, Japan, Cat No. RR82LR). For the primers, HBV 379F (5'-GTG TCT GCG GCG TTT TAT CA-3') and HBV 476R (5'-GAC AAA CGG GCA ACA TAC CTT-3') were used and PCR was conducted with a Rotor-Gene qPCR machine (Qiagen, ver 2.1.0).

## Western blot analysis

Cell lysis was achieved using a lysis buffer composed of 50 mM Tris–HCl (pH 8.0), 0.1% SDS, 150 mM NaCl, and 1% NP-40, complemented with protease inhibitors (Roche, Basel, Switzerland, Cat No. 11836153001). Cell lysates underwent separation through SDS-PAGE and subsequent transfer onto a nitrocellulose membrane (Amersham Bio-science, Amersham, Cat No. 10600003). Membranes were treated with 5% skim milk (BD DIFCO, NJ, USA) or 5% BSA (Bovogen, East Keilor, Australia, Cat No. BsAs 0.1) in PBS-T (PBS with 0.1% Tween 20) for one hour at room temperature to mitigate non-specific antibody binding. Membranes were incubated with appropriate antibodies to HA, p53, HBs, HBc, DNA methyltransferases 1 (DNMT1), DNMT3a, DNMT3b, γ-tubulin (Santa Cruz Biotechnology, Cat No. sc-7392, sc-126, sc-17787, 1:500 dilution), HBx (Millipore, Cat No. MAB8419, 1:2000 dilution), E6AP (Thermo Fisher Scientific, Cat No. PA3-843, 1:2000 dilution), seven in absentia homolog 1 (Siah-1) (Abcam, Cambridge, UK, Cat No. ab2237, 1:2000 dilution). Following this step, the membranes underwent successive incubation with a suitable anti-mouse secondary antibody conjugated with HRP, (Bio-Rad, Hercules, CA, USA, Cat No. BR170-6516, 1:3000 dilution), anti-goat IgG (H+L)-HRP (Thermo Scientific Scientific, Cat No. 31400, 1:10,000 dilution), or anti-rabbit IgG (H+L)-HRP (Bio-Rad, Cat No. BR170-6515, 1:3000 dilution). An ECL kit (Advansta, San Jose, CA, USA, Cat No. K-12043-D20) was used to detect the protein bands on the membrane using the ChemiDoc XRS imaging system (Bio-Rad).

## Immunoprecipitation

Immunoprecipitation (IP) assay was conducted using the Classic Magnetic IP/Co-IP kit (Thermo Fisher Scientific, Cat No. 88804). In brief, cell lysates were subjected to overnight incubation with an anti-HBx antibody (Millipore, Burlington, MA, USA Cat No. 8419) at 4°C to promote the generation of immune complexes, whole-cell lysates from $4 \times 10^5$ cells per 60 mm-diameter plate were either infected with HBV or transiently transfected with the specified expression plasmids based on the indicated condition. Subsequently, the immune complexes were collected by incubation with Protein A/G magnetic beads (0.25 mg) after washing. The lysates were further incubated for an additional hour, and the beads were isolated using a magnetic stand (Pierce). Finally, the eluted antigen/antibody complexes underwent western blotting utilizing the specified antibodies.

## Immunofluorescence analysis

A double-label indirect immunofluorescence assay (IFA) was executed following established procedures [35]. Cells were cultured on coverslips and fixed with 4% formaldehyde at 20°C for 15 minutes and permeabilized with methanol at −20°C for 10 minutes. Then cells were subjected to overnight incubation with an appropriate antibody at 4°C, followed by subsequent incubation with anti-mouse IgG-FITC (Sigma-Aldrich, Cat No. F0257-1ML, 1:100 dilution)

and anti-rabbit IgG-rhodamine (Invitrogen, Waltham, MA, USA, Cat No. 31670; 1:200 dilution) at room temperature for an hour. Prepared slides were mounted using UltraCruz mounting medium (Santa Cruz Biotechnology) and observed through an Eclipse fluorescence microscope (Nikon, Tokyo, Japan). Densitometric analysis of the immunofluorescence signal was conducted using ImageJ software (NIH, Bethesda, MD, USA, ver 6.0.1).

## Methylation-specific PCR (MSP)

Utilizing the QIAamp DNA Mini Kit (Qiagen), cellular genomic DNA extraction was carried out. Following this, 1 μg of genomin cDNA underwent bisulfite modification using the EpiTect Bisulfite kit (Qiagen), following the manufacturer's protocols. The modified DNA (100 ng) underwent E6AP MSP, employing a methylated primer pair, E6AP-Me-1F, and E6AP-Me-1R, as well as an unmethylated primer pair, E6AP-Un-1F, and E6AP-Un-1R, as previously outlined [36].

## Cell viability assay

MTT assay was conducted to assess cell viability, in accordance with previously established protocols [37]. In brief, cells were incubated in 96-well plates at a density of $10^4$ cells per well. Subsequently, cells were treated with μM 3-(4,5-dimethylthiazol-2-yl)-2,5-diphenyltetrazolium bromide (MTT, Sigma-Aldrich, Cat No. M2128) for 4 hours at 37˚C incubator. Levels of MTT were assessed by measuring the absorbance at 550 nm after the formazan compounds, produced from MTT by the mitochondrial dehydrogenases of viable cells, were dissolved with DMSO (Sigma-Aldrich).

## Statistical analysis

The provided data represents the average ± standard deviation obtained from a minimum of three independent experiments. Statistical analyses were performed using a two-tailed Student's t-test. Results with a P value greater than 0.05 were considered not statistically significant, whereas those with a P value of 0.05 or less were regarded as statistically significant.

## Results

### ATRA suppresses HBV replication in a p53-dependent manner

First, we explored whether ATRA exhibits distinct effects on HBV replication in human hepatocellular carcinoma cells based on the p53 status. HepG2 cells express WT p53, while Hep3B cells do not, which provides a distinctive platform for conducting parallel and comparative analysis of the contributions of p53 to HBV-related molecular mechanisms [38]. Exposure to HBV particles derived from a 1.2-mer HBV replicon [39], using a recently optimized HBV infection protocol [31], was conducted on HepG2-NTCP and Hep3B-NTCP cells, which stably express the HBV receptor NTCP [40]. HBV replication in HepG2-NTCP and Hep3B-NTCP cells was confirmed by western blot analysis of viral proteins (HBx, HBsAg, and HBcAg) in cell lysates (Fig 1A and 1D) and by measuring virus particles in culture supernatants through qPCR (Fig 1B and 1E). Two forms of HBsAg, large (L)- and middle (M)-HBsAg were specifically detected in the infected cells, whereas the antibody used in this study failed to detect small (S)-HBsAg in the same cell extracts (Fig 1A and 1D), as demonstrated previously [33]. These findings suggest observable levels of efficient HBV replication in both HepG2-NTCP and Hep3B-NTCP cells under the specified experimental conditions.

The treatment of ATRA led to a dose-dependent reduction in the levels of both intracellular HBV proteins and extracellular secreted virus particles during the process of HBV replication

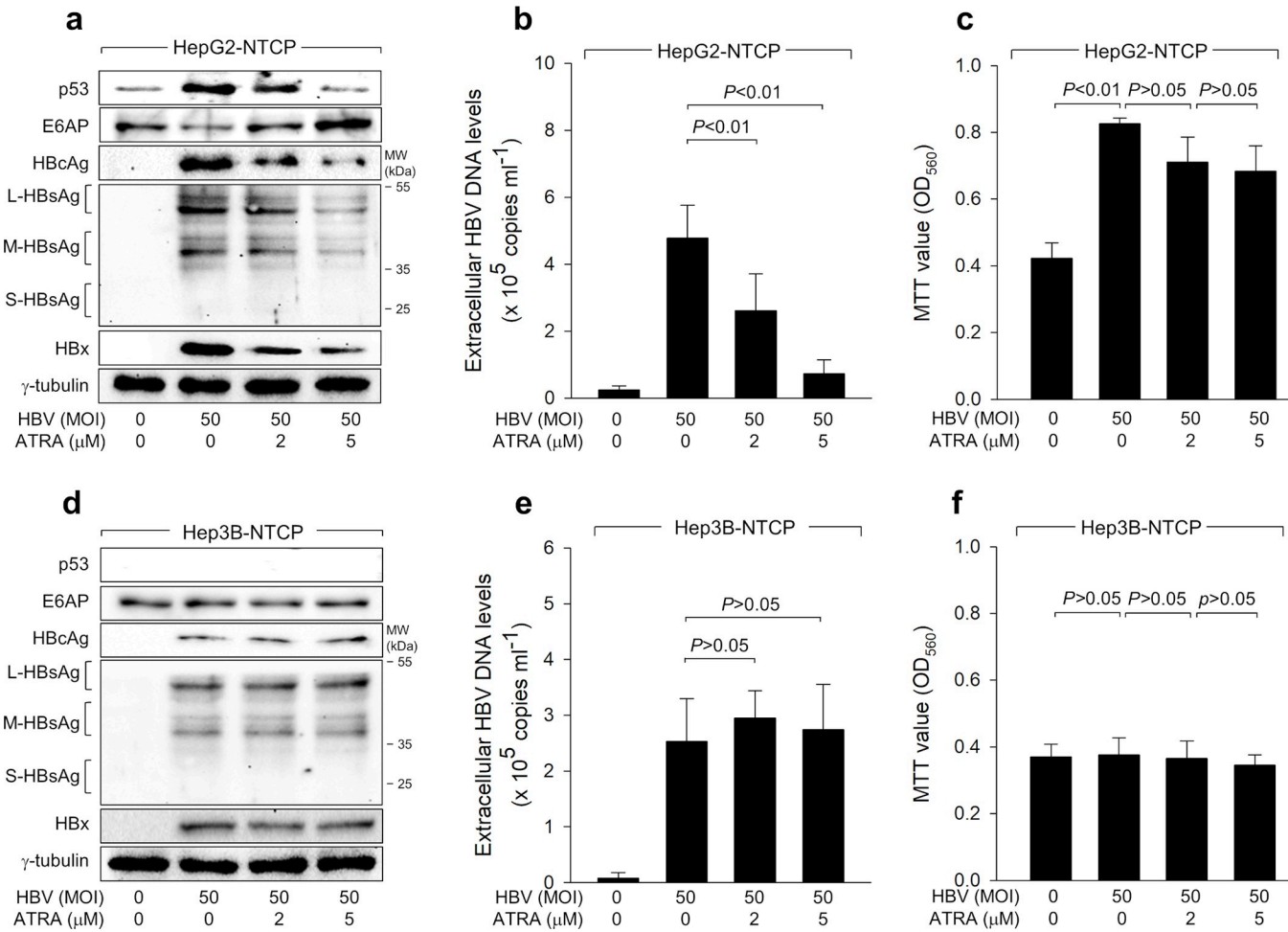

**Fig 1. ATRA suppresses HBV replication in a p53-dependent manner.** Both Hep3B-NTCP and HepG2-NTCP cells were subjected to HBV infection for 24 h at 50 MOI, followed by two washes with serum-free DMEM. Subsequently, cells were further incubated for an extra 3 days in DMEM including 4% PEG 8000, 3% FBS, and 2% DMSO. Treatment with the specified ATRA concentrations was administered 24 h before harvesting. (a, d) Western blotting of cell lysates was conducted to assess the levels of the suggested proteins. (b, e) The HBV particle levels released from the cells from (a, d) were quantified using quantitative real-time PCR (qPCR). The means ± standard deviation (SD) presented here are derived from four independent experiments (n = 4). (c, f) Cell viability was determined by the MTT assay (n = 9).

in HepG2-NTCP cells (Fig 1A and 1B) but not in Hep3B-NTCP cells, where p53 was absent (Fig 1D and 1E). To exclude the possibility that the inhibition of HBV replication by ATRA is a consequence of its cytotoxic effects, MTT assays were performed to examine whether ATRA affects cell viability under our experimental conditions. HBV infection increased the MTT value in HepG2-NTCP cells but not in Hep3B-NTCP cells, indicating that HBV infection stimulates cell growth in a p53-dependent manner, as previously shown [11, 24]. Treatment with ATRA significantly decreased the MTT value in HepG2-NTCP cells infected with HBV but not in Hep3B-NTCP cells (Fig 1C and 1F), indicating that this effect is also p53-dependent. However, the value obtained from the HepG2-NTCP cells infected with HBV and treated with 5 μM ATRA was still higher than that from the control cells (Fig 1C). Furthermore, ATRA failed to induce the downregulation of MTT values in HepG2-NTCP and Hep3B-NTCP cells infected with HBx-null HBV (Fig 2C), indicating that HBx is essential for this effect. Therefore, we conclude that the inhibition of HBV replication by ATRA in HepG2-NTCP cells is not due to the cytotoxic effect but rather a p53- and HBx-dependent effect of ATRA.

## ATRA downregulates HBx protein levels in a p53-dependent manner

Subsequently, we explored the mechanism through which ATRA hinders HBV replication in a manner dependent on p53. Interestingly, ATRA failed to downregulate the levels of intracellular HBV proteins and extracellular HBV DNA levels in HepG2-NTCP and Hep3B-NTCP cells infected with HBx-null HBV (Fig 2A and 2B). Furthermore, ATRA diminished the levels of HBx ectopically expressed in HepG2 cells, while no such effect was detected in Hep3B cells (Fig 2E and 2F). These findings indicate that ATRA reduces protein levels of HBx which is known to positively regulate HBV replication [13, 17, 18], thereby inhibiting HBV replication in human hepatocellular carcinoma cells in a p53-dependent mechanism. To confirm that ATRA downregulates HBx levels in a p53-dependent manner, we attempted to diminish p53 expression in HepG2 cells and express ectopic p53 in Hep3B cells in the presence of both HBx and ATRA. Certainly, the knockdown of p53 in HepG2 cells significantly abolished the ability of ATRA to downregulate HBx levels, while the introduction of ectopic p53 expression in Hep3B cells facilitated ATRA-induced downregulation of HBx levels (Fig 2G and 2H). Taken together, our findings lead to the conclusion that ATRA downregulates HBx protein levels, thereby inhibiting HBV replication in human hepatoma cells in a manner dependent on p53.

## ATRA-mediated E6AP upregulation suppresses HBx in a p53-dependent manner

Next, our investigation focused on the mechanism by which ATRA downregulates HBx levels in a p53-dependent manner. Previous studies have demonstrated the primary role of the Ub-proteasome system in determining HBx levels [11, 35]. Two E3 ligases, Siah-1 and E6AP, are involved in the Ub-dependent proteasomal degradation of HBx [35, 41, 42]. Consistent with previous reports [35, 42], in HepG2 cells, HBx upregulated both p53 and Siah-1 levels but downregulated E6AP levels, whereas these effects were not observed in Hep3B cells (Fig 2E and 2F). In the absence of HBx, ATRA upregulated p53 and Siah-1 levels in HepG2 cells in a dose-dependent manner (Fig 2D).

ATRA also upregulated p53 and Siah-1 levels at low concentrations (below 1.0 μM), but downregulated their levels at high concentrations (over 2.0 μM) in HepG2 cells expressing HBx (Fig 2E). On the contrary, ATRA similarly regulated E6AP levels in the presence and absence of HBx, although the effects were also dependent on the concentrations of ATRA: ATRA downregulated E6AP levels at low concentrations (below 1.0 μM) but upregulated E6AP levels at high concentrations (over 2.0 μM) (Fig 2D and 2E). Siah-1, rather than E6AP, seems to play a crucial role in the downregulation of HBx levels by ATRA at low concentrations as previously demonstrated [11], if considering that Siah-1 levels were upregulated whereas E6AP levels were slightly downregulated or unaffected under the condition (Fig 2E). However, E6AP is likely to show a dominant role in the downregulation of HBx levels under the condition that ATRA at high concentrations upregulated E6AP levels but downregulated Siah-1 levels in HepG2 cells. Therefore, Siah-1 and E6AP may play a distinct role in the proteasomal degradation of HBx depending on the concentration of ATRA. For the following experiments, 5 μM concentration was selected to focus on the role of E6AP in the ATRA-mediated inhibition of HBV replication.

Contrary to HepG2 cells, ATRA did not impact E6AP levels during ectopic HBx expression in Hep3B cells (Fig 2F) and HBV replication in Hep3B-NTCP cells (Fig 1D). ATRA also could not regulate levels of Siah-1 in Hep3B cells expressing HBx (Fig 2F). Additionally, p53 knockdown downregulated E6AP and Siah-1 levels under the influence of ATRA, resulting in the upregulation of HBx levels (Fig 2G). Moreover, ectopic p53 expression in Hep3B cells enabled ATRA to regulate E6AP, Siah-1, and HBx levels (Fig 2H), as demonstrated in HepG2 cells

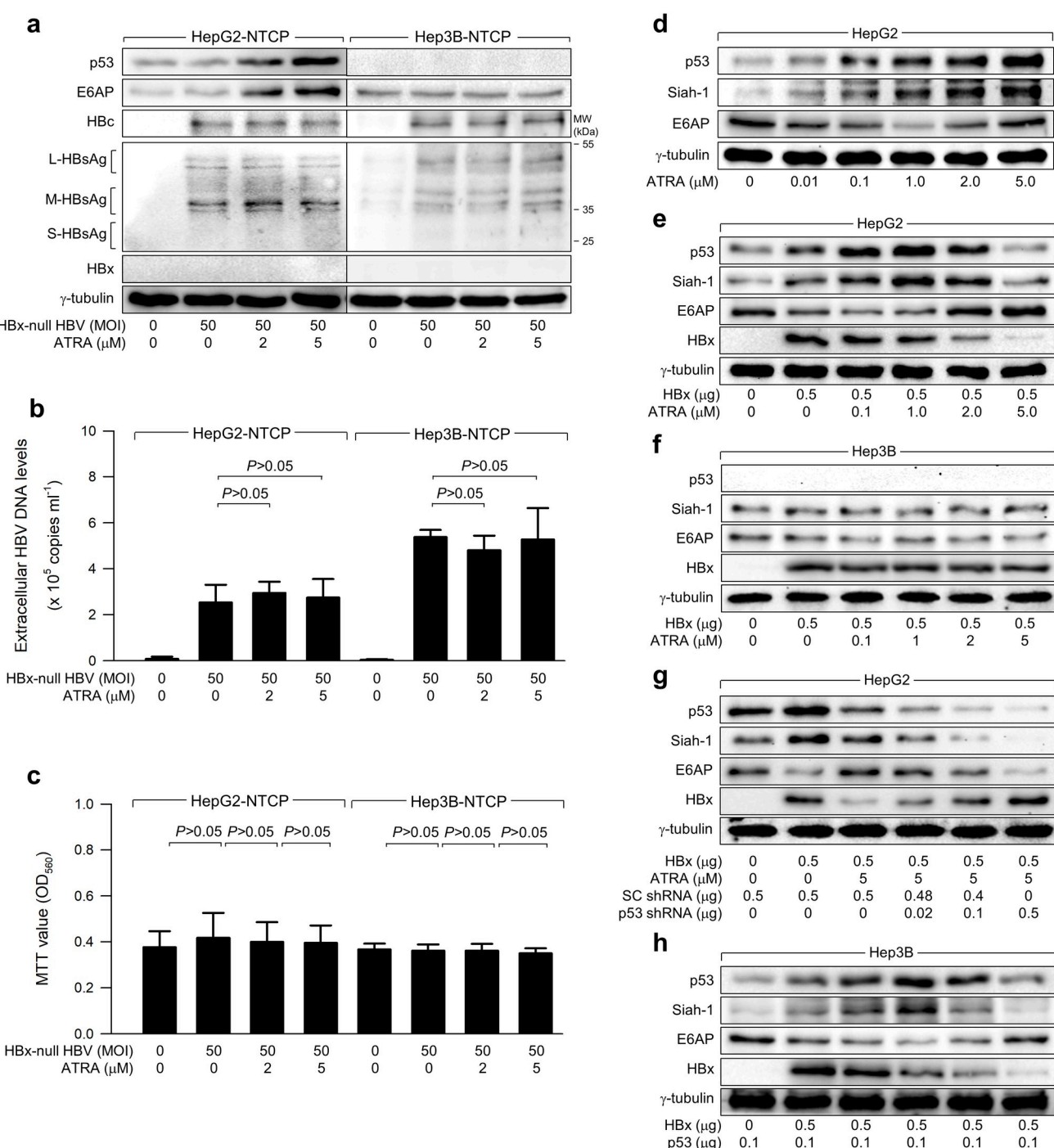

**Fig 2. ATRA downregulates HBx levels in a p53-dependent manner.** (a) HepG2-NTCP and Hep3B-NTCP cells were infected with HBx-null HBV and treated with ATRA as in Fig 1A, followed by western blotting. (b) The levels of HBV particles from the cells prepared in (a) were determined by qPCR (n = 6). (c) The viability of the cells prepared in (a) was measured using the MTT assay (n = 9). (d to h) Either HepG2 or Hep3B cells were transfected with the suggested plasmids for 24 h and then treated with ATRA at the described concentrations for an additional 24 h before harvesting, followed by western blotting.

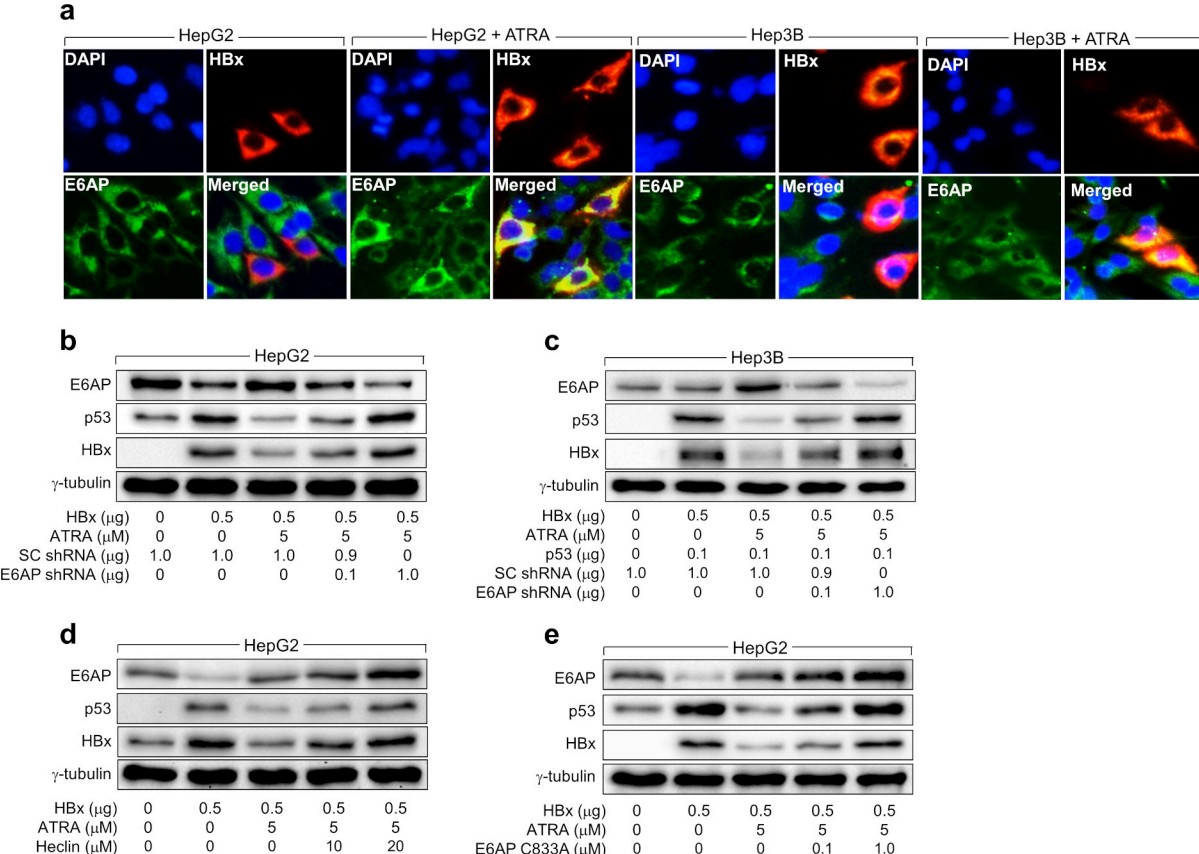

**Fig 3. ATRA-mediated E6AP upregulation suppresses HBx in a p53-dependent manner.** (a) HepG2 and Hep3B cells, cultured on coverslips, were transfected with an HBx expression plasmid for 24 h and subsequently either mock-treated or treated with 5 μM ATRA for an additional 24 h. The cells were then processed for double-label indirect immunofluorescence, incubating with anti-HBx monoclonal and anti-E6AP antibodies, followed by incubation with anti-mouse IgG–FITC and anti-rabbit IgG–rhodamine antibodies to visualize HBx (red) and E6AP protein (green), respectively. Nuclei (blue) were stained with 4′,6-diamidino-2-phenylindole (DAPI). (b to e) Either HepG2 or Hep3B cells were transfected with the indicated plasmids for 24 h and then treated with ATRA at the indicated concentrations for an additional 24 h before harvesting, followed by western blotting. For (d), cells were treated with Heclin at the indicated concentrations for 24 h before harvesting.

(Fig 2E). The p53-dependent upregulation of E6AP levels by ATRA was also investigated by IFA. E6AP and HBx were colocalized primarily in the cytoplasm of both HepG2 and Hep3B cells (Fig 3A). The signal from E6AP was weakened by ectopic HBx expression in HepG2 cells but unaffected in Hep3B cells. Treatment with ATRA increased E6AP levels in HepG2 cells expressing HBx, whereas this effect was not detected in Hep3B cells expressing HBx (Fig 3A). Therefore, the data from the IFA also indicate that ATRA increases E6AP levels to decrease HBx levels in a p53-dependent manner.

To validate the involvement of E6AP in the ATRA-induced downregulation of HBx protein levels in the presence of p53, we attempted to knock down E6AP in HepG2 cells while expressing HBx with the treatment of ATRA. The downregulation of E6AP levels led to the restoration of HBx levels and subsequent upregulation of p53 levels in ATRA-treated HepG2 and Hep3B cells expressing ectopic p53 (Fig 3B and 3C). Additionally, we utilized Heclin, a specific inhibitor of HECT-type E3 ligases [43], to investigate whether the E3 ligase activity of E6AP is required for ATRA-induced HBx downregulation. Indeed, Heclin abolished the effect of ATRA on HBx levels in a dose-dependent manner, resulting in the upregulation of p53 levels

in HepG2 cells in the presence of ATRA (Fig 3D). Furthermore, the ectopic expression of E6AP C833A, featuring a cysteine-to-alanine substitution at the active site and unable to form a thioester with Ub [44, 45], also prevented ATRA from downregulating HBx levels, resulting in the upregulation of p53 levels in ATRA-treated HepG2 (Fig 3E). These results definitively demonstrate that the E3 ligase activity of E6AP plays a crucial role in the ATRA-mediated downregulation of HBx levels in human hepatoma cells.

## ATRA activates E6AP expression via promoter hypomethylation in a p53-dependent manner

Our next focus was to unveil the mechanism through which ATRA upregulates E6AP levels in the presence of p53 and HBx. E6AP expression is primarily regulated via DNA methylation of the CpG islands located on the promoter [31, 46]. In agreement with a prior study [31], HBx elevated the levels of DNMT1, 3a, and 3b, leading to promoter hypermethylation of E6AP and subsequently causing the downregulation of E6AP levels in HepG2 cells (Fig 4A and 4B). Interestingly, treatment with ATRA downregulated DNMT1, 3a, and 3b levels and induced promoter hypomethylation of E6AP, irrespective of the presence of HBx, resulting in the E6AP protein level upregulation in HepG2 cells (Fig 4A and 4B). None of these effects were evident in Hep3B cells (Fig 4A and 4B), indicating that p53 is required not only for the activation of the cellular DNA methylation system by HBx but also for its abolishment by ATRA. Based on these findings, we conclude that ATRA activates E6AP expression via promoter hypomethylation in a p53-dependent manner.

## ATRA increases HBx degradation via E6AP-mediated ubiquitination to suppress HBV replication in a p53-dependent manner

Having confirmed that ATRA downregulates HBx levels by increasing E6AP expression, we further explored whether ATRA reduces the protein stability of HBx in human hepatoma cells in a manner dependent on p53. To assess this, we treated cells with CHX to further block protein synthesis while monitoring the levels of HBx and γ-tubulin in HepG2 and Hep3B cells (Fig 5A). Notably, the half-life ($t_{1/2}$) of HBx in HepG2 cells was determined to be 62.2 min, but it was drastically shortened to 30.2 min by treatment with ATRA, indicating the ability of ATRA to reduce HBx stability. The half-life of HBx was higher in Hep3B cells ($t_{1/2}$ = 82.9 min), which do not express p53, a negative regulator of HBx stability [35, 47]. In contrast to HepG2 cells, ATRA did not exhibit a discernible effect on the half-life of HBx in Hep3B cells ($t_{1/2}$ = 80.8 min) (Fig 5A), further emphasizing the role of p53 in the regulation of HBx levels by ATRA in human hepatoma cells.

To further investigate the participation of E6AP in the ATRA-induced downregulation of HBx levels, we explored whether ATRA enhances the p53-dependent E6AP-mediated ubiquitination of HBx. To address this, we introduced HBx and HA-tagged Ub into both HepG2 and Hep3B cells, followed by immunoprecipitation of the Ub-complexed HBx, with or without ATRA treatment. Co-IP data demonstrated the interaction of both E6AP and Siah-1 with HBx, resulting in its ubiquitination, as evidenced by the presence of different sizes of smeared polyubiquitinated HBx bands in HepG2 cells (Fig 5B, lane 1). ATRA enhanced the protein interaction between HBx and E6AP but diminished the protein interaction between Siah-1 and HBx, presumably because ATRA upregulated E6AP levels while downregulating Siah-1 levels in HepG2 cells (Fig 5B, lane 2). As ATRA induced powerful ubiquitination of HBx and subsequent reduction of HBx protein levels, E6AP rather than Siah-1 appeared to play a critical role in the ubiquitination and proteasomal degradation of HBx under our experimental conditions. These findings are consistent with the observed decrease in the interaction between HBx

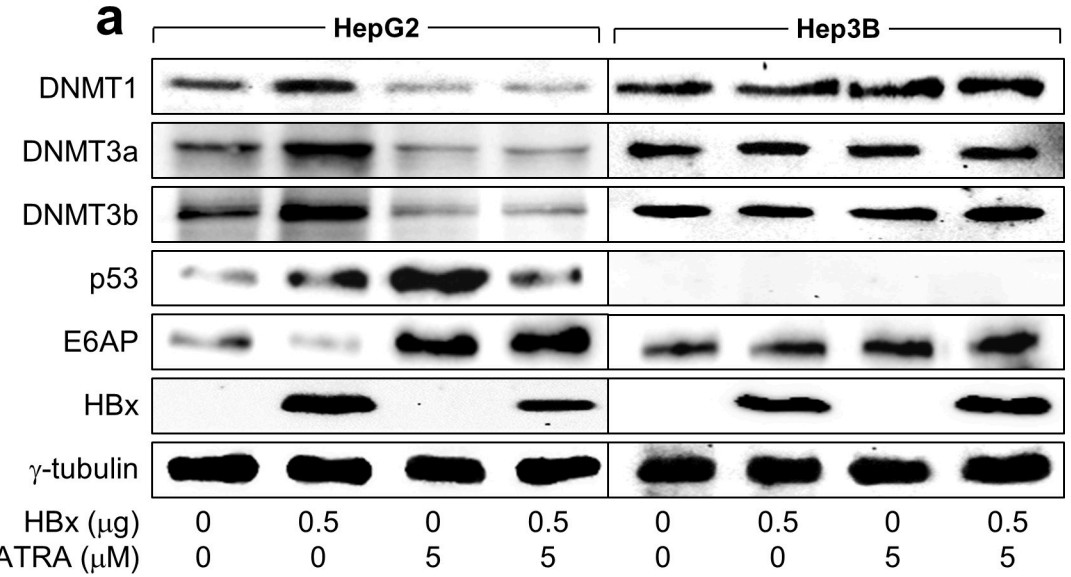

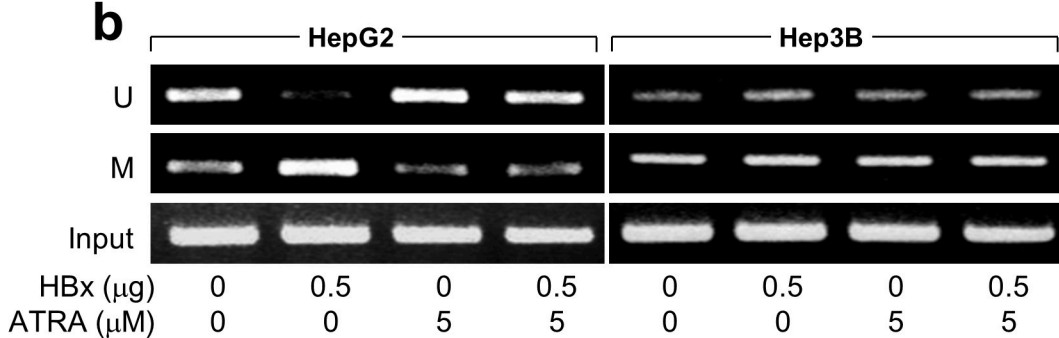

**Fig 4. ATRA activates E6AP expression via promoter hypomethylation in a p53-dependent manner.** After 24 hours of transfection with the specified amounts of HBx expression plasmid, both HepG2 and Hep3B cells were treated with ATRA for an additional 24 hours. (a) The levels of the indicated proteins were assessed by western blotting. (b) Methylation-specific PCR (MSP) was conducted to determine the methylation status of CpG sites within the E6AP promoter, distinguishing between unmethylated (U) and methylated (M) states.

and E6AP upon E6AP depletion in the presence of ATRA in HepG2 cells. Consequently, there was a notable increase in HBx protein levels due to a significant reduction in HBx ubiquitination (Fig 5B, lane 4). Moreover, the administration of the peptide-aldehyde proteasome inhibitor MG132 significantly attenuated the capacity of ATRA to downregulate HBx levels. As a result, the HBx protein levels were equalized in the presence and absence of ATRA (Fig 5C). These findings confirm that ATRA downregulates HBx levels through E6AP by proteasomal degradation.

## ATRA suppresses HBV replication via E6AP-mediated proteasomal degradation of HBx in a p53-dependent manner

Finally, we examined whether ATRA, in a p53-dependent manner, induces E6AP-mediated proteasomal degradation of HBx to repress HBV replication in human hepatocellular carcinoma cells. Consistent with data acquired from the HBx overexpression system (Fig 5), the

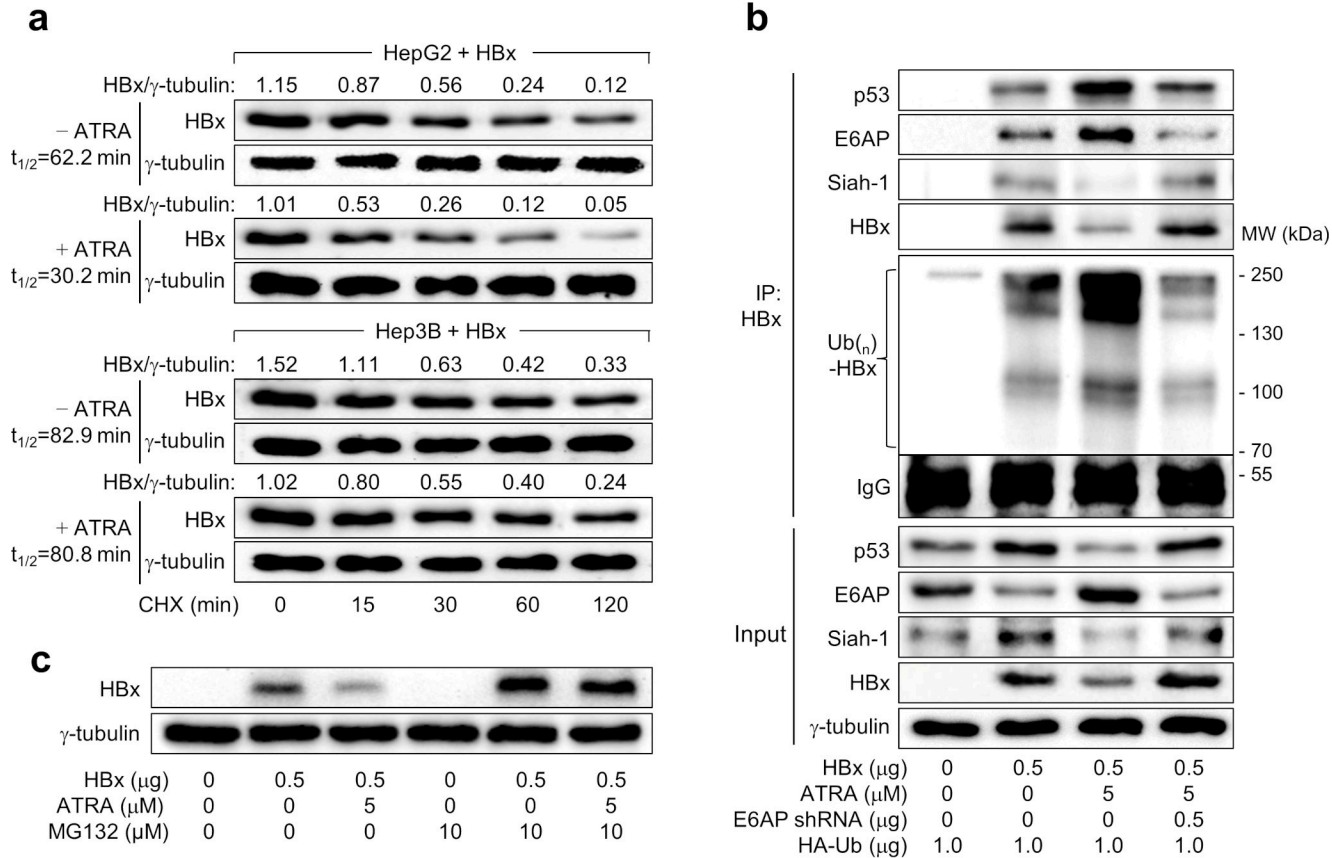

**Fig 5. ATRA increases HBx degradation via E6AP-mediated ubiquitination to suppress HBV replication in a p53-dependent manner.** (a) HepG2 and Hep3B cells were transiently transfected with HA-HBx and treated with ATRA as in Fig 2E and then dosed with 50 μM of protein synthesis inhibitor cycloheximide (CHX) for the presented time before harvesting, then followed by western blotting. Using Image J 1.53k image analysis software (NIH), γ-tubulin and HBx levels were quantified. The calculation of half-life ($t_{1/2}$) of HBx was conducted based on the levels of HBx compared to the loading control (γ-tubulin). (b) HepG2 cells were transiently transfected with the indicated plasmids and then ATRA treatment was conducted as in Fig 2E. Immunoprecipitation was conducted using an anti-HBx antibody to isolate total HBx protein from cell lysates. Subsequently, western blot analysis was performed using suitable antibodies to detect p53, E6AP, Siah-1, HBx, and HA-Ub-complexed HBx. Levels of the specified proteins in the cell lysates are depicted in the input lane. (c) HepG2 cells were transiently transfected with a plasmid containing HBx gene and treated with ATRA as in Fig 2E. (c) Cells prepared following the protocol in Fig 2E were subjected to either mock treatment or treatment with MG132 for 4 h before cell harvest, and subsequent western blotting was performed.

negative effect of ATRA on the half-life of HBx during HBV replication was much higher in HepG2-NTCP cells (67.9 min to 48. 9 min) than in Hep3B-NTCP cells (103.9 min to 99.4 min). Furthermore, in the context of HBV infection in HepG2-NTCP cells, ATRA strengthened the protein interaction between E6AP and p53, resulting in substantial ubiquitination of HBx and subsequent degradation of HBx protein levels (Fig 6B). Furthermore, the knockdown of E6AP not only diminished the interaction between HBx and E6AP but also reduced the ubiquitination of HBx in the presence of ATRA, resulting in the subsequent increase of HBx levels in HepG2-NTCP cells throughout HBV infection (Fig 6B). Correspondingly in HepG2-NTCP cells, E6AP knockdown promoted HBV replication in the presence of ATRA, with no discernible impact detected on Hep3B-NTCP cells, as evidenced by an augmentation in the levels of intracellular HBV proteins and extracellular HBV DNA levels (Fig 6C-6H). To determine whether E6AP is critical for the regulation of HBx even at low concentrations of ATRA, we either knocked down or overexpressed E6AP in conditions of low ATRA concentrations. Interestingly, the knockdown of E6AP inhibited HBV replication in the presence of 1 μM

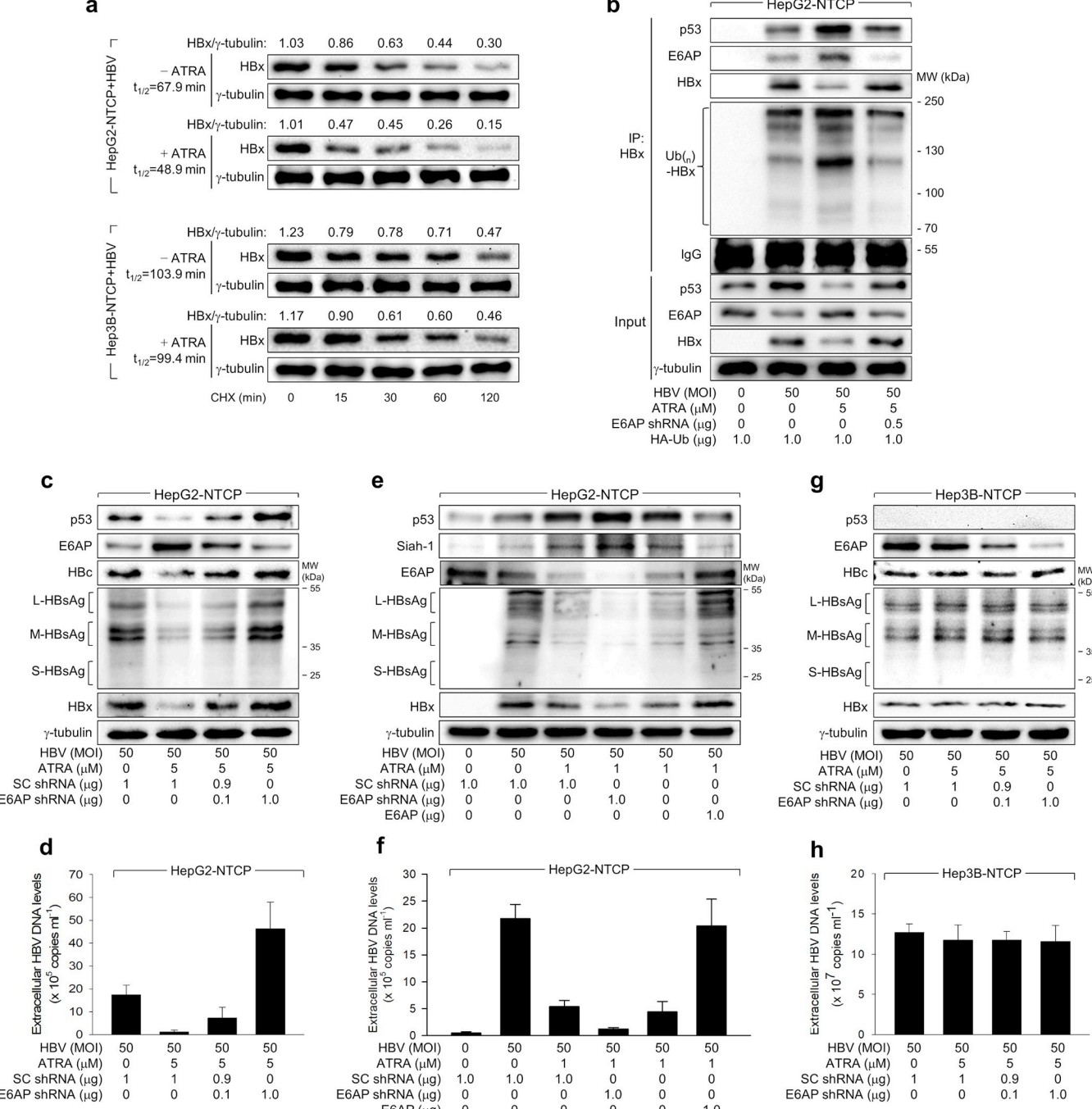

**Fig 6. ATRA suppresses HBV replication via E6AP-mediated proteasomal degradation of HBx in a p53-dependent manner.** (a) HepG2-NTCP and Hep3B-NTCP cells were infected with HBV and treated with 5 μM ATRA as in Fig 1A and 1B Cells were then treated with 50 μM CHX to determine the $t_{1/2}$ values of HBx as in Fig 5A. (b) HepG2-NTCP cells transfected with the indicated plasmids for 24 h were infected with HBV and treated with ATRA as in Fig 1A. Co-immunoprecipitation-coupled western blotting was performed as in Fig 5A. (c,e,g) HepG2-NTCP and Hep3B NTCP cells were infected with HBV and treated with ATRA as in (b), followed by western blotting. (d,f,h) Levels of HBV particles in the culture supernatants from cells in (c,e,g) were measured by qPCR (n = 4).

ATRA because it upregulated p53 and Siah-1 levels under the condition (Fig 6E and 6F). In addition, E6AP overexpression stimulated HBV replication in the presence of 1 μM ATRA through the downregulation of p53 and Siah-1 levels (Fig 6E and 6F). These results support the data shown in a previous report [11] and Fig 2E demonstrating the dominant role of Siah-1 over E6AP in the determination of HBx levels at low concentrations of ATRA. In summary, these results support the conclusion that ATRA suppresses HBV replication through E6AP-mediated proteasomal degradation of HBx in a p53-dependent manner.

## Discussion

Previous reports have demonstrated that ATRA inhibits the replication of diverse viruses, including HBV [6–12]. However, the detailed action mechanisms of ATRA against these viruses remain unclear. This study illustrated that ATRA at pharmacological concentrations (2–5 μM), already shown to induce the differentiation of APL cells in patients [48] and suppress HBV replication in primary human hepatocytes [10], inhibits HBV replication in cultured human hepatoma cells. Our findings also suggest that ATRA depends on the interactions between two host factors (p53 and E6AP) and a viral factor (HBx) to execute its anti-viral potential during HBV replication in human hepatoma cells.

Several reports have provided evidence that HBx functions as a positive regulator of HBV replication, either by stimulating HBV mRNA synthesis [13, 17, 18] or by deregulating cellular signaling pathways [49, 50]. The current study presents several lines of evidence supporting the concept that ATRA regulates HBV replication by reducing the levels of HBx through E6AP-mediated proteasomal degradation. First, ATRA upregulated E6AP levels during HBV infection in human hepatoma cells (Fig 1A). Second, ATRA failed to inhibit *in vitro* replication of HBx-null HBV (Fig 2A and 2B). Third, ATRA increased E6AP-mediated ubiquitination and proteasomal degradation of HBx in both HBx overexpression and *in vitro* HBV replication systems (Figs 5B and 6B). Fourth, all the anti-HBV effects of ATRA were abolished when E6AP was knocked down (Figs 3 and 6), resulting in a recovery of HBV replication in the presence of ATRA.

Previous reports have demonstrated that both ATRA and HBx upregulate p53 levels but through different mechanisms. HBx upregulates p53 levels through the activation of the ATM-Chk2 pathway [19, 25, 42, 51], while ATRA increases p53 levels through the activation of the p14-MDM2 pathway [21]. Consistently, ATRA and HBx individually upregulated p53 levels in HepG2 cells (Fig 2D and 2E). According to the findings in the present study, the effects of ATRA on p53 levels in the presence of HBx can be opposite depending on its concentrations: ATRA upregulated p53 levels at low concentrations (below 1 μM), whereas it downregulated p53 levels at high concentrations (over 2 μM) (Fig 2E). It is not easy to explain how ATRA differently regulates p53 levels in the presence of HBx depending on the concentrations. One possible explanation is the presence of a negative feedback regulatory circuit, which may enable HBx to maintain p53 levels within a certain range during HBV replication. For example, HBx may facilitate the E6AP-mediated degradation of p53, as demonstrated with the E6 protein of HPV-16 and -18 [52], notably under the condition that ATRA elevates E6AP and p53 levels. Indeed, the knockdown of E6AP upregulated p53 levels under the condition that ATRA at 5 μM upregulated E6AP levels (Fig 3B), establishing an inverse correlation between p53 and E6AP.

It is unknown how ATRA upregulates E6AP levels in the presence of HBx. Interestingly, a negative correlation between p53 and E6AP levels was consistently observed in the presence of HBx (Figs 1A, 2E, and 2H). Interestingly, the knockdown of p53 converted it into a positive one by downregulating E6AP levels in the presence of HBx and ATRA (Fig 2G). Additionally,

in the absence of HBx, the correlation between p53 and E6AP was differentially affected by ATRA in a concentration-dependent manner (Fig 2A). Furthermore, both ATRA and HBx failed to affect E6AP levels in the absence of p53 (Fig 2F). Therefore, p53 should be a major determinant in the regulation of E6AP levels in the presence of ATRA and HBx, although the detailed mechanism remains a major subject of further investigation.

E6AP expression is known to be primarily regulated through DNA methylation [46]. In addition, an antagonism between HBx and ATRA for the regulation of the cellular DNA methylation system has been reported. According to a recent report [31], HBx activates the cellular DNA methylation system to inhibit E6AP expression via promoter hypermethylation during HBV replication. Consistently, HBx increased levels of DNMT1, 3a, and 3b and inhibited E6AP expression via promoter hypermethylation in HepG2 but not in Hep3B cells (Fig 4), emphasizing the p53-dependent nature of this effect. Conversely, ATRA decreased the levels of DNMTs, irrespective of HBx presence, in a condition that resulted in the reduction of p53 levels. This led to the promoter hypomethylation of E6AP and subsequent upregulation of its protein levels. This aligns with a previous study that showed ATRA activates p14 expression through promoter hypomethylation in the presence of HBx [25]. ATRA likely overcomes the stimulatory effect of HBx on the DNA methylation system under our experimental conditions.

Both Siah-1 and E6AP are known to act as E3 ligases responsible for the ubiquitination and proteasomal degradation of HBx [35, 42]. Although Siah-1 does not require p53 for its E3 ligase activity, its expression is activated by p53 through a p53-binding site in the promoter [42, 53]. Consistently, Siah-1 levels were constantly proportional to p53 levels in the presence of ATRA and HBx (Fig 2D and 2E). On the other hand, both E6AP expression and its enzyme activity are regulated by p53 [35]. Consistently, p53 was required not only for the regulation of E6AP levels by ATRA (Fig 2D and 2E) but also for the E6AP-mediated degradation of HBx in the presence of ATRA (Fig 6C and 6E). In the absence of p53, ATRA had minimal effects on E6AP, Siah-1, and HBx levels in human hepatoma cells (Fig 2F and 2G). Therefore, both Siah-1 and E6AP are likely to be involved in the p53-dependent downregulation of HBx levels by ATRA.

According to data from the co-IP experiments, ATRA decreased the interaction between Siah-1 and HBx under the condition that it increased the interaction between E6AP and HBx in HepG2 cells (Figs 5B and 6B). Additionally, the interaction between Siah-1 and HBx increased as E6AP knockdown decreased the interaction between E6AP and HBx in the presence of ATRA (Fig 6B). Therefore, E6AP and Siah-1 may antagonize each other, presumably via competition, for binding to HBx. As Siah-1 and E6AP levels were differentially regulated by ATRA in a concentration-dependent manner (Figs 2E and 6E), they may play distinct roles in the regulation of HBx levels depending on the concentrations of ATRA. Indeed, Siah-1 has been proven to play a major role in the regulation of HBx levels by low concentrations of ATRA (below 1 μM) [11], whereas the present study showed that E6AP plays a crucial role in the degradation of HBx by high concentrations of ATRA (over 2 μM) (Figs 3 and 6E). Considering the physiological concentration of ATRA (0.2 μM) [54], Siah-1 may govern HBx degradation to control HBV replication in patients as part of the host innate defense systems. However, E6AP may substitute Siah-1 to act as a major regulator of HBx levels in HBV-positive patients who are administered pharmacological concentrations of ATRA (2–5 μM) [48] for the treatment of cancers, virus infections, and other diseases.

Taken together, our study emphasizes the distinct roles of Siah-1 and E6AP in the regulation of HBx stability, particularly in the context of p53-mediated mechanisms. These findings provide valuable insights into the intricate regulatory network that governs HBV replication and highlight the potential therapeutic implications of ATRA in the management of HBV-related pathologies, particularly hepatocellular carcinoma. Further exploration of the precise

molecular mechanisms underlying the interplay between ATRA, p53, Siah-1, and E6AP may offer promising avenues for the development of novel therapeutic strategies targeting HBV-associated diseases.

## Supporting information

**S1 Raw image.**
(PDF)

## Acknowledgments

We thank W.-S. Ryu for providing the HBV replicon system used in this study.

## Author Contributions

**Conceptualization:** Jiwoo Han, Kyung Lib Jang.

**Data curation:** Jiwoo Han, Kyung Lib Jang.

**Formal analysis:** Jiwoo Han.

**Funding acquisition:** Kyung Lib Jang.

**Investigation:** Jiwoo Han.

**Methodology:** Jiwoo Han.

**Project administration:** Kyung Lib Jang.

**Software:** Jiwoo Han.

**Supervision:** Kyung Lib Jang.

**Validation:** Jiwoo Han, Kyung Lib Jang.

**Writing – original draft:** Jiwoo Han.

**Writing – review & editing:** Kyung Lib Jang.

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
