## [Decision Letter · Decision Letter 0]

25 Mar 2024

PONE-D-24-01455All-trans retinoic acid downregulates HBx levels via E6-associated protein-mediated proteasomal degradation to suppress hepatitis B virus replicationPLOS ONE

Dear Dr. Jang,

Thank you for submitting your manuscript to PLOS ONE. After careful consideration, we feel that it has merit but does not fully meet PLOS ONE’s publication criteria as it currently stands. Therefore, we invite you to submit a revised version of the manuscript that addresses the points raised during the review process.

We look forward to receiving your revised manuscript.

Kind regards,

Kishor Pant

Academic Editor

PLOS ONE

Reviewers' comments:

Reviewer's Responses to Questions

**Comments to the Author**

1. Is the manuscript technically sound, and do the data support the conclusions?

Reviewer #1: Yes

Reviewer #2: Yes

2. Has the statistical analysis been performed appropriately and rigorously? 

Reviewer #1: Yes

Reviewer #2: Yes

3. Have the authors made all data underlying the findings in their manuscript fully available?

Reviewer #1: Yes

Reviewer #2: Yes

4. Is the manuscript presented in an intelligible fashion and written in standard English?

Reviewer #1: Yes

Reviewer #2: Yes

5. Review Comments to the Author

Reviewer #1: Based on the study presented in this manuscript, I would like to give following comments

-Manuscript looks good and written in standard English

-Data are presented in scientific manner.

- Please mention the catalog number of all kit or reagent used in this study in materials and methods section.

Reviewer #2: The study seems to be very interesting that uncovers some of the previous unanswered questions. However, there are few concerns about the study that have been raised in the reviewer's comments and need to addressed.

6. PLOS authors have the option to publish the peer review history of their article (what does this mean?). If published, this will include your full peer review and any attached files.

Reviewer #1: No

Reviewer #2: **Yes: **AMIT KUMAR MISHRA

---

## [Author Response · Author response to Decision Letter 0]

12 Apr 2024

We have attached file named "Response to Reviewrs commments".

File includes these contents below.

Comments:

1. Result 2 shows that expression of p53, Siah-1 and E6AP varies depending upon the concentration of ATRA used in the experiment. Why 5mM concentration was selected and not the lower concentration for the study from fig 3 onwards.

Answer: As observed in Figures 2d and 2e, the concentration of ATRA appeared to inversely correlate with the levels of HBx, showcasing a continuous decrease as ATRA concentration increased. In association with this trend, while low concentrations of ATRA led to an upregulation in Siah-1 levels, higher concentrations marked an increase in the levels of E6AP. Our previous study has shown that the increase in Siah-1 at lower concentrations of ATRA contributed to the reduction of HBx [1]. However, at higher concentrations, where Siah-1 levels were observed to decline, the attention was shifted towards another E3 ligase, E6AP, whose levels were found to increase with rising concentrations of ATRA. This led to the hypothesis that E6AP might play a role in diminishing HBx levels, prompting further experiments to investigate the potential involvement of E6AP in the degradation of HBx. Furthermore, upon treatment with various concentrations of ATRA, the optimal concentration that maximized both the decrease in HBx levels and the increase in E6AP levels, while also considering cell toxicity, was determined to be 5 μM. This concentration was subsequently applied in the experiments.

Also please refer to lines 292 to 294.

2. HBx protein has been shown to be required for successful HBV infection and replication. In the absence of HBx, only basal level HBV replication is expected. In result 2 (a), authors have shown that ATRA failed to diminish the HBV protein expression in HepG2 cells infected with HBx null HBV and have concluded that HBx is required for inhibitory effect of ATRA. It would be better to include the same experimental data in Hep3B cells (Effect of ATRA on HBV proteins in Hep3B cells infected with HBx null HBV) to make it more relatable for the readers.

Answer: We newly performed HBx-null HBV experiments in Hep3B-NTCP cells as shown in Fig 2a to 2c.

3. In result 3, is there an explanation for ATRA downregulating E6AP levels at low concentrations and upregulating at higher concentrations. What is the reason for its reverse effect (ATRA increases p53 and Siah-1 level at low concentration and decreases at higher concentration) on expression of p53 and Siah-1 levels.

Answer: Our previous reports and data in Fig 4 show that E6AP expression is primarily regulated via DNA methylation. However, it is unknown how ATRA downregulates E6AP levels at low concentrations but upregulates them at higher concentrations. Considering that ATRA in the presence and absence of HBx minimally affected both E6AP promoter methylation and protein levels in Hep3B cells, the possible involvement of p53 in the opposite regulation of E6AP promoter methylation depending on its protein levels can be speculated. In addition, it has been well established that p53 is degraded by E6AP in the presence of E6 of human papillomavirus type 16 (HPV-16) and HPV-18. Therefore, it is possible to speculate that a similar mechanism may exist where p53 is degraded by E6AP in the presence of HBx. As p53 is a positive regulator for Siah-1, protein levels of Siah-1 follow the level of p53 in the high concentration of ATRA. 

Please refer to lines 437 to 457.

4. Result 5 title does not match with the results explained. The data and the explanation say that ATRA “increases” HBx degradation via E6AP mediated ubiquitination and thus suppresses HBV replication. However, the title of the figure says that ATRA “downregulates” p53-dependent HBx degradation via E6AP ubiquitination to suppress HBV replication. 

Answer: The title was changed to “ATRA increases HBx degradation via E6AP-mediated ubiquitination to suppress HBV replication in a p53-dependent manner”

5. In result 5, the data has been produced at 5mM ATRA concentration. It would be interesting to know that how does lower ATRA concentration affects E6AP mediated ubiquitination and HBx degradation? If results vary compared to effects shown at 5mM ATRA concentration?

Answer: We performed new experiments to show whether knockdown or overexpression of E6AP could affect HBV replication at low ATRA concentration (1 μM) as presented in Figures 6e and f. Interestingly knockdown of E6AP inhibited HBV replication probably because it upregulated p53 and Siah-1 levels. In addition, E6AP overexpression stimulated HBV replication because it downregulated p53 and Siah-1 levels. These results support the data shown in Figure 2e demonstrating the dominant role of Siah-1 over E6AP in the determination of HBx levels at low ATRA concentrations.

Please refer to lines 389 to 397.

6. HepG2.2.15 cells are stably HBV expressing lines and were designed to study the HBV infection, replication, and life cycle. It would be advisable to replicate the key findings from this paper (ATRA treatment increases E6AP which in turn ubiquitinates HBx to suppress HBV replication) in this cell line.

Answer: Referencing the paper "Michailidis E, Pabon J, Xiang K, Park P, Ramanan V, Hoffmann H-H, et al. A robust cell culture system supporting the complete life cycle of hepatitis B virus. Scientific Reports. 2017;7(1):16616.", we have established an HBV infection system. This system has enabled us to conduct more accurate experiments on HBV replication. Our newly modified HBV infection system addresses these limitations by incorporating a cell line that expresses the NTCP receptor, allowing for the study of the complete HBV life cycle, including viral entry, replication, and egress, in a manner that closely mimics natural infection. This comprehensive model provides a more robust platform for the evaluation of antiviral drugs, the investigation of viral pathogenesis, and the study of HBV-induced carcinogenesis, offering significant advantages over the HepG2.2.15 cell line and marking a step forward in HBV research. 

HBV infection experiments using the HepG2.2.15 cell line used to be useful for studying certain aspects of HBV replication and expression but face several limitations in several key areas [2]:

1) Lack of Natural Infection Dynamics: HBV DNA in HepG2.2.15 cells is chromosomally integrated, allowing for the study of viral replication but not the initial stages of infection, such as viral entry. This limitation restricts our understanding of the early events in HBV infection, which are crucial for developing targeted interventions.

2) Inability to Model Direct Infection: HepG2.2.15 cells lack the sodium taurocholate co-transporting polypeptide (NTCP), a critical receptor for HBV entry into hepatocytes. This absence makes these cells insusceptible to infection from HBV, thereby limiting the scope of infection models and the study of natural viral entry mechanisms.

3) Inadequacy for Studying Viral Entry and Uncoating: Critical early stages of the HBV life cycle, such as adsorption, cellular entry, and uncoating, cannot be investigated using HepG2.2.15 cells. These processes are essential for understanding HBV infectivity and for developing entry inhibitors as potential therapeutic agents.

Considering these limitations, we are of the view that employing our newly established HBV infection system presents a more logical approach than continuing with the HepG2.2.15 system.

7. Figure legend 2(a), It says “HepG2-NRCP” rather than “HepG2-NTCP”. Please correct that.

Answer: Correction completed

References

1. Han J, Jang KL. All-trans Retinoic Acid Inhibits Hepatitis B Virus Replication by Downregulating HBx Levels via Siah-1-Mediated Proteasomal Degradation. Viruses. 2023;15(7). Epub 20230627. doi: 10.3390/v15071456. PubMed PMID: 37515144; PubMed Central PMCID: PMCPMC10386411.

2. Xu R, Hu P, Li Y, Tian A, Li J, Zhu C. Advances in HBV infection and replication systems in vitro. Virol J. 2021;18(1):105. Epub 20210529. doi: 10.1186/s12985-021-01580-6. PubMed PMID: 34051803; PubMed Central PMCID: PMCPMC8164799.

---

## [Decision Letter · Decision Letter 1]

29 May 2024

All-trans retinoic acid downregulates HBx levels via E6-associated protein-mediated proteasomal degradation to suppress hepatitis B virus replication

PONE-D-24-01455R1

Dear Dr. Jang,

We’re pleased to inform you that your manuscript has been judged scientifically suitable for publication and will be formally accepted for publication once it meets all outstanding technical requirements.

Kind regards,

Kishor Pant

Academic Editor

PLOS ONE

Additional Editor Comments (optional):

Reviewers' comments:

Reviewer's Responses to Questions

**Comments to the Author**

1. If the authors have adequately addressed your comments raised in a previous round of review and you feel that this manuscript is now acceptable for publication, you may indicate that here to bypass the “Comments to the Author” section, enter your conflict of interest statement in the “Confidential to Editor” section, and submit your "Accept" recommendation.

Reviewer #1: All comments have been addressed

Reviewer #2: All comments have been addressed

2. Is the manuscript technically sound, and do the data support the conclusions?

Reviewer #1: Yes

Reviewer #2: Yes

3. Has the statistical analysis been performed appropriately and rigorously? 

Reviewer #1: Yes

Reviewer #2: Yes

4. Have the authors made all data underlying the findings in their manuscript fully available?

Reviewer #1: Yes

Reviewer #2: Yes

5. Is the manuscript presented in an intelligible fashion and written in standard English?

Reviewer #1: Yes

Reviewer #2: Yes

6. Review Comments to the Author

Reviewer #1: (No Response)

Reviewer #2: I feel satisfied with the authors response and the manuscript looks in perfect shape to be published.

7. PLOS authors have the option to publish the peer review history of their article (what does this mean?). If published, this will include your full peer review and any attached files.

Reviewer #1: No

Reviewer #2: No

---

## [Editor Report · Acceptance letter]

31 May 2024

PONE-D-24-01455R1 

PLOS ONE

Dear Dr. Jang, 

I'm pleased to inform you that your manuscript has been deemed suitable for publication in PLOS ONE. Congratulations! Your manuscript is now being handed over to our production team.

Kind regards, 

on behalf of

Dr. Kishor Pant 

Academic Editor

PLOS ONE